# Molecular Identification and Characterization of Two Groups of Phytoplasma and *Candidatus* Liberibacter Asiaticus in Single or Mixed Infection of *Citrus maxima* on Hainan Island of China

**DOI:** 10.3390/biology11060869

**Published:** 2022-06-06

**Authors:** Shao-Shuai Yu, An-Na Zhu, Wei-Wei Song, Wei Yan

**Affiliations:** 1Coconut Research Institute, Chinese Academy of Tropical Agricultural Sciences, Wenchang 571339, China; zan810037317@163.com (A.-N.Z.); songweiwei426@sohu.com (W.-W.S.); ywei123@catas.cn (W.Y.); 2College of Forestry, Hainan University, Haikou 570228, China

**Keywords:** phytoplasma, *Candidatus* Liberibacter asiaticus, *Citrus maxima*, molecular identification, mixed infection

## Abstract

**Simple Summary:**

Based on the 16S rRNA and β-operon gene fragments, two subgroups of phytoplasma—CmPII-hn belonging to 16SrII-V and CmPXXXII-hn belonging to 16SrXXXII-D—and *Candidatus* Liberibacter asiaticus CmLas-hn were detected separately in 12, 2 and 6 out of 54 citrus samples of *Citrus maxima*, an important economic crop in Hainan Island, China, infected with Huanglongbing. Among the detection results, mixed infection of 16SrII-V subgroup phytoplasma and *Candidatus* Liberibacter asiaticus was identified in four samples, accounting for 7.4%. The CmPII-hn strain was in a cluster belonging to the 16SrII-V subgroup, with a 99% bootstrap value. The CmPXXXII-hn strain, *Trema tomentosa* witches’ broom phytoplasma, belonging to 16SrXXXII-D, and the other 16SrXXXII subgroup strains were in one cluster with a 99% bootstrap value. Sixteen variable loci were detected in the 16S rRNA genes of the tested 16SrXXXII group phytoplasma strains, of which two bases had an insertion/deletion. The CmLas-hn strain and *Candidatus* Liberibacter asiaticus were in one independent cluster with a 99% bootstrap value. In the study, *Citrus maxima*, showing yellowing and mottled leaves as disease symptoms, were found, which could have been infected separately by 16SrII-V and 16SrXXXII-D subgroup phytoplasmas or could have been subjected to mixed infection by 16SrII-V phytoplasmas and *Candidatus* Liberibacter asiaticus in China.

**Abstract:**

The pathogens associated with citrus Huanglongbing symptoms, including yellowing and mottled leaves in *Citrus maxima*, an important economic crop on Hainan Island of China, were identified and characterized. In the study, detection, genetic variation and phylogenetic relationship analysis of the pathogens were performed based on 16S rRNA and β-operon gene fragments specific to phytoplasma and *Candidatus* Liberibacter asiaticus. The results indicated that the pathogens—such as phytoplasma strains of CmPII-hn belonging to the 16SrII-V subgroup and CmPXXXII-hn belonging to the 16SrXXXII-D subgroup, as well as *Candidatus* Liberibacter asiaticus strains CmLas-hn—were identified in the diseased plant samples, with numbers of 12, 2 and 6 out of 54, respectively. Among them, mixed infection with the 16SrII-V subgroup phytoplasma and *Candidatus* Liberibacter asiaticus was found in the study, accounting for 7.4% (four samples). The phytoplasma strains of CmPII-hn—*Tephrosia purpurea* witches’ broom, *Melochia corchorifolia* witches’ broom and *Emilia sonchifolia* witches’ broom—were clustered into one clade belonging to the 16SrII-V subgroup, with a 99% bootstrap value. The phytoplasma strains of CmPXXXII-hn and *Trema tomentosa* witches’ broom belonging to 16SrXXXII-D, and the other 16SrXXXII subgroup strains were clustered into one clade belonging to the 16SrXXXII group with a 99% bootstrap value. There were 16 variable loci in the 16S rRNA gene sequences of the tested 16SrXXXII group phytoplasma strains, of which two bases had an insertion/deletion. The strains of *Candidatus* Liberibacter asiaticus, identified in the study and the strains that had been deposited in GenBank, were in one independent cluster with a 99% bootstrap value. To our knowledge, this is the first report showing that *Citrus maxima* can be infected by 16SrII-V and16SrXXXII-D subgroup phytoplasmas in China. Moreover, this is also the first report in which the plants are co-infected by 16SrII-V subgroup phytoplasmas and *Candidatus* Liberibacter asiaticus. More comprehensive and detailed identification and characterization of the pathogens associated with the diseased symptoms in *Citrus maxima* on the island in China would be beneficial for epidemic monitoring and for the effective prevention and control of related plant diseases.

## 1. Introduction

Phytoplasmas are a large group of cell-wall-less and phloem-limited plant pathogenic prokaryotes that infect nearly a thousand plant species and cause serious economic losses around the world [1,2,3]. Under natural conditions, the prokaryotic pathogens can be transmitted by insects and grafting, manipulating the parasitized plants and insects by secreting pathogenic effectors [4,5,6,7,8,9]. After infecting the plant hosts, the phytoplasmas trigger the related plant pathological changes, such as chloroplast degradation and spatiotemporal redistribution of phytohormones [10,11]. This leads to the related plant diseases, with the symptoms of witches’ broom, virescence, phyllody, cluster top, internode shortening, leaflets, yellowing, dwarfing, crinkling and plant growth decline and even death [4,5,6,7,10,11]. Phytoplasmas cannot be cultured in vitro. Therefore, molecular analyses of DNA sequences from the phytoplasmas are more important for their detection, identification and classification [12,13,14,15]. The 16S rRNA gene sequences of the phytoplasmas are mainly used for the molecular identification and characterization of pathogens [12,13,14,15]. Based on extensive RFLP and phylogenetic analyses of the 16S rRNA gene sequence, 20 different clades (groups) of phytoplasmas were systematically recognized by Seemüller et al. [16]. Based on the computer-simulated RFLP analysis of 16S rRNA gene sequences from several hundred phytoplasma strains, 28 16Sr group phytoplasmas have been classified by Wei et al. [13]. Thus far, 33 16Sr groups of phytoplasmas have been classified on the basis of RFLP analyses and/or sequencing of the 16S rRNA gene sequences [5].

Citrus Huanglongbing is one of the most destructive citrus diseases affecting citrus production in China and around the world [17,18,19,20,21,22]. Taxonomically, there are three different species of *Candidatus* Liberibacter spp. that have been reportedly associated with Citrus Huanglongbing, namely *Candidatus* Liberibacter asiaticus, *Candidatus* Liberibacter africanus and *Candidatus* Liberibacter americanus, which were categorized based on their presumptive geographical origins (Asian, African and American continents, respectively) and 16S rDNA molecular classification [17,21,22,23]. *Candidatus* Liberibacter spp. are phloem-restricted plant pathogens and are easily transmitted through the citrus psyllid [17,23]. Citrus crops infected by the pathogens of *Candidatus* Liberibacter spp. could show disease symptoms of yellow shoots, mottled leaves and even tree death [17,19]. The plant diseases caused by the phytoplasma and the *Candidatus* Liberibacter spp. are extremely difficult to eradicate due to their rapid spread, causing the related economic plants to lose their productive capacity [5,17]. Therefore, it is very important for the epidemic monitoring and the control management of the disease to identify the pathogens and types of disease as systematically and in as much detail as possible.

Plants showing disease symptoms under natural conditions may be caused by a single or mixed infection by two or more plant pathogens. A large number of studies have shown that phytoplasma could co-infect plants, particularly phytoplasmas belonging to the different 16Sr subgroups or other pathogens, such as *Candidatus* Liberibacter spp. [22,24,25,26]. There have been reports that citrus crops showing Huanglongbing symptoms were mixed infected by phytoplasma and the *Candidatus* Liberibacter spp. throughout the world [25,26]. *Candidatus* Liberibacter spp. contains three species, which are pathogens of Citrus Huanglongbing disease, causing disease symptoms of yellowing and mottled leaves. Citrus Huanglongbing could affect citrus plants, such as *Citrus maxima*, *Citrus sinensis*, *Citrus aurantifolia* and *Citrus*×*limon* [21,23,25,26]. Previous studies indicated that the Citrus Huanglongbing affecting the citrus industry in China was only associated with *Candidatus* Liberibacter asiaticus. Until now, there have been no reports that other *Candidatus* Liberibacter spp. strains, such as *Candidatus* Liberibacter africanus and *Candidatus* Liberibacter americanus, have been identified in citrus plant samples with Citrus Huanglongbing symptoms in China [17,23].

The plant *Citrus maxima* is an important economic crop and featured fruit in Hainan Province, a tropical island in China. Disease symptoms in the plant, including yellowing and mottled leaves, were found on Hainan Island in China, causing serious economic losses and affecting the local citrus industry. The disease symptoms of *Citrus maxima* are similar to the symptoms caused by Citrus Huanglongbing disease. Furthermore, some phytoplasma diseases were reported previously in the regions associated with the disease symptoms of *Citrus maxima* [27,28,29,30]. Therefore, the plants with disease symptoms were suspected to be associated with the pathogens of phytoplasma and *Candidatus* Liberibacter asiaticus, based on a field investigation and a previous study [17,18,19,23,27,28,29,30]. At present, there are no reports of mixed infection with phytoplasma and *Candidatus* Liberibacter asiaticus on the citrus plant hosts and even other plant hosts on Hainan Island in China. Therefore, the detection and identification of the phytoplasmas and *Candidatus* Liberibacter asiaticus from the diseased *Citrus maxima* samples were simultaneously carried out in this study. The pathogens associated with the disease symptoms of *Citrus maxima* were clarified in the study, which could be helpful for more targeted and effective prevention and control of *Citrus maxima* diseases.

## 2. Materials and Methods

### 2.1. Plant Samples and DNA Preparation

The plants of *Citrus maxima* that displayed disease symptoms of yellowing and mottled leaves, as shown in Figure 1, are suspected to be infected by phytoplasma or *Candidatus* Liberibacter asiaticus, were found in Ding’an county in Hainan Province in China during 2020 to 2021. The symptoms seriously impacted their growth and caused severe financial losses. The yellow and crinkled leaves of the diseased plant samples were collected on a farm in Ding’an county with the coordinate positions of 110°8′59.36″ E, 19°34′33.73″ N. The healthy *Citrus maxima* samples were also collected in the same locations as a control. The sampling site and related locations in Hainan Province, a tropical island off the southern coast of China, are shown in Figure 2. Economic palm crops, such as areca palm, coconut and oil palm, were also planted on the farm. Moreover, there were many phytoplasma-infected plants reported previously, such as *Melochia corchorifolia*, *Emilia sonchifolia*, *Trema tomentosa* and *Waltheria indica*, which were also found on the farm [27,28,29,30]. Total DNA of the diseased plant samples was extracted based on the CTAB method, according to the rapid extraction kit for plant genomic DNA (Tiangen Biotech Co., Ltd., Beijing, China) [31]. In particular, 0.01 g of fresh symptomatic plant tissue containing leaf and leaf midribs from these plant materials was used for the DNA extraction. The DNA samples were stored at −20 °C until use.

### 2.2. PCR Amplification and DNA Sequencing

Based on the 16S rRNA gene, the primer pairs of R16mF2/R16mR2 [12,32] were used for the detection of the phytoplasmas. Based on the gene fragments of the 16S rRNA and β-operon, the primer pairs of OI1/OI2c [33] and A2/J5 [34] were used for the detection of *Candidatus* Liberibacter asiaticus. The 25 μL amplification reaction was used for the phytoplasma detection, including a 20 ng DNA template, 0.5 μL R16mF2/R16mR1 (10 μM), 12.5 μL 2× PCR premix (0.05 U μL^−1^ *Taq* DNA polymerase, 4 mmol L^−1^ MgCl_2_, and 0.4 mmol L^−1^ dNTPs) and ddH_2_O added to generate 25 μL. The reaction conditions were as follows: 94 °C for 5 min; 94 °C for 45 s, 52 °C for 1 min 30 s and 72 °C for 1 min for 35 cycles in total; 72 °C for 10 min for the final extension. The 25 μL amplification reaction was also used for the detection of *Candidatus* Liberibacter asiaticus. The PCR components were similar to the R16mF2/R16mR1 PCR, except that the primer pairs OI1/OI2c and A2/J5 were used. The reaction conditions of the primer pairs OI1/OI2c and A2/J5 were as follows: 94 °C for 5 min; 94 °C for 1 min, 56 °C for 1 min and 72 °C for 2 min for 35 cycles in total; 72 °C for 10 min for the final extension. The PCR products were visually detected by 1.0 % (*w*/*v*) agarose gel electrophoresis through ethidium bromide straining. The PCR products were purified and sequenced using an ABI PRISM 3700 automated DNA sequencer (Sangon Biotech Co., Ltd., Guangzhou, China).

### 2.3. Sequence Analysis

The nucleotide sequences obtained in the study were assembled and edited using DNAMAN software, version 6.0 (Lynnon Corporation, Vaudreuil-Dorion, Quebec, QC, Canada) and the EditSeq program (DNAStar package, Madison, WI, USA). All nucleotide sequences were aligned based on the algorithm of pairwise alignment by performing an analysis of multiple sequence alignment using DNAMAN 6.0. Homology matrices of the DNA sequences of the pathogen strains were displayed by carrying out the multiple sequence alignment method, revealing the sequence similarity. The nucleotide sequences were compared and analyzed to identify the pathogen strains using the Basic Local Alignment Search Tool (BLAST) (https://blast.ncbi.nlm.nih.gov/Blast.cgi (accessed on 22 March 2022)) depending on the sequence databases. Based on the 16S rRNA gene sequences, the phytoplasmas were identified and classified with the interactive online phytoplasma classification tool, *i*PhyClassifier [35]. Phylogenetic analysis was performed with MEGA software, version 7.0 [36], employing the neighbor-joining (NJ) method with a 1000 bootstrap value [37]. The reference phytoplasma strains, such as onion yellow phytoplasma OY-M (GenBank accession: AP006628), aster yellow phytoplasma AYWB (CP000061), Australia grape yellow phytoplasma AUSGY (AM422018), strawberry lethal yellow phytoplasma SLY (CP002548) and apple proliferation phytoplasma AP (CU469464), were used as outgroups in the study. The 16S rRNA gene sequence of the *Trema tomentosa* witches’ broom phytoplasma strain (MW138004), with the length of 1303 bp, was selected and utilized as a putative consensus sequence. The polymorphic site positions of the analyzed genes were numbered corresponding to the putative consensus gene sequence, and the single-nucleotide polymorphisms (SNP) of the genes in the tested phytoplasma strains are listed in Table 1.

### 2.4. Nucleotide Sequence Accession Numbers

The DNA sequences identified in the study were all deposited in the GenBank database using the Portal (GenBank) tool for submission of the ribosomal RNA (rRNA) sequences (https://submit.ncbi.nlm.nih.gov/about/genbank/ (accessed on 28 March and 6 April 2022)) and the Bankit tool for submission of the protein-coding genes (https://submit.ncbi.nlm.nih.gov/about/bankit/ (accessed on 28 March 2022)). The GenBank accession numbers of the DNA sequences identified in the study were obtained and are listed in the study. The DNA sequences were shown in Appendix A.

## 3. Results

### 3.1. Molecular Detection and Identification of the Pathogens Associated with the Disease Symptoms in Citrus Maxima

The target-amplified bands of the 16S rRNA and β-operon gene fragments specific to phytoplasmas and *Candidatus* Liberibacter asiaticus were detected in the DNA of the diseased samples of *Citrus maxima* that displayed symptomatic symptoms, whereas they were not found in the symptomless sample DNA of the plant. After sequencing and editing, there were two types of 16S rRNA gene sequences of phytoplasma associated with the *Citrus maxima* disease symptoms. The BLAST search, based on the 16S rRNA genes of the phytoplasmas, indicated that one type of 16S rRNA gene of the *Citrus maxima* phytoplasma had 100% sequence identity (with 100% coverage) with the 16S rRNA genes of the 16SrII peanut witches’ broom (PnWB) phytoplasma group members, such as the phytoplasma strains of *Melochia corchorifolia* witches’ broom (MZ353520), *Tephrosia purpurea* witches’ broom (MW616560) and *Emilia sonchifolia* witches’ broom (MW353971). Moreover, another type of 16S rRNA gene of the *Citrus maxima* phytoplasma had a 100% sequence identity (with 100% coverage) with the 16S rRNA genes of the 16SrXXXII Malaysian periwinkle virescence (MaPV) phytoplasma group members, such as the phytoplasma strain of *Trema tomentosa* witches’ broom (MW138004). The phytoplasma strains belonging to the 16SrII group associated with the *Citrus maxima* disease were identical and described as CmPII-hn, with a length of 1316 bp (ON159857), and the phytoplasma strains belonging to the 16SrXXXII group associated with the *Citrus maxima* disease were identical and described as CmPXXXII-hn, with a length of 1317 bp (ON159856). The 16S rRNA gene sequences obtained in the study that were specific to *Candidatus* Liberibacter asiaticus were identical, with a length of 1010 bp (ON080846). The BLAST search, based on the 16S rRNA gene sequences of *Candidatus* Liberibacter asiaticus, indicated that the 16S rRNA genes from the DNA of the diseased *Citrus maxima* samples had 100% sequence identity (with 100% coverage) with those of *Candidatus* Liberibacter asiaticus, such as the strains CP010804, MK142763 and MG384791. The *Candidatus* Liberibacter asiaticus strains associated with the *Citrus maxima* disease in the study were described as CmLas-hn. The β-operon encoding gene sequences obtained in the study that were specific to *Candidatus* Liberibacter asiaticus were identical, with a length of 581 bp (ON098932). The BLAST search, based on the β-operon encoding gene sequences of *Candidatus* Liberibacter spp., indicated that the β-operon encoding gene sequences from the DNA of the diseased *Citrus maxima* samples in the study had a 100% sequence identity (with 100% coverage) with those of *Candidatus* Liberibacter asiaticus, such as the strains CP010804, MK542517 and MG418842. Among the detected pathogens in the 54 *Citrus maxima* diseased samples, 16SrII-V subgroup phytoplasma strains were detected in twelve samples, accounting for 22.2%; 16SrXXXII-D subgroup phytoplasma strains were detected in two samples, accounting for 3.7%; and *Candidatus* Liberibacter asiaticus strains were detected in six samples, accounting for 11.1%. Among them, mixed infection with 16SrII-V subgroup phytoplasma and *Candidatus* Liberibacter asiaticus was identified in four samples, accounting for 7.4%.

### 3.2. Virtual RFLP Analysis of the Citrus Maxima Phytoplasma Strains

The virtual RFLP analysis of the phytoplasma strains associated with the *Citrus maxima* disease was performed using an interactive online tool, *i*PhyClassifier [35], and was based on the computer-simulated digestions of 17 restriction enzymes: *Alu*I, *BamH*I, *Bfa*I, *Bst*UI, *Dra*I, *Eco*RI, *Hae*III, *Hha*I, *Hinf*I, *Hpa*I, *Hpa*II, *Kpn*I, *Sau*3AI, *Mse*I, *Rsa*I, *Ssp*I and *Taq*I. The results obtained through the *i*PhyClassifier operations indicated that the virtual RFLP pattern derived from the 16S rRNA gene sequence fragment of the CmPII-hn phytoplasma strain was identical (similarity coefficient 1.00) to the reference pattern of 16Sr group II, subgroup V (KY568717). Therefore, the CmPII-hn phytoplasma strain under study is a member of the 16SrII-V subgroup. Based on the *i*PhyClassifier operations, the virtual RFLP pattern derived from the 16S rRNA gene sequence fragment of the CmPXXXII-hn phytoplasma strain was identical (similarity coefficient 1.00) to the reference pattern of 16Sr group XXXII, subgroup D (MW138004). Therefore, the CmPXXXII-hn phytoplasma strain under study is a member of the 16SrXXXII-D subgroup, a phytoplasma taxonomic subgroup that was first reported in China [29]. Thus far, there are two phytoplasmas belonging to the 16SrXXXII-D subgroup that have been identified: one is the *Citrus maxima* phytoplasma strain CmPXXXII-hn, which was found in Ding’an county in Hainan Island, China, in the study; the other is the *Trema tomentosa* witches’ broom phytoplasma strain (MW138004), which was found in the Ding’an and Qionghai counties on the island, which was reported previously [29]. The virtual RFLP profiles of the 16S rRNA gene fragments of the *Citrus maxima* phytoplasma strains CmPII-hn and CmPXXXII-hn, obtained from *i*PhyClassifier, are shown in Figure 3.

### 3.3. Genetic Variation Characteristics of the Phytoplasma and the Candidatus Liberibacter Asiaticus Infecting Citrus Maxima

The analysis of the multiple sequence alignment indicated that the CmPII-hn phytoplasma strains were closely related to the phytoplasma strains, such as *Tephrosia purpurea* WB-hn (MW616560), *Melochia corchorifolia* WB-hnda (MZ353520), *Emilia sonchifolia* WB-hnda (MW353971) and Peanut WB-tw (JX403944). There was no base variation site in the 16S rRNA gene sequences of the phytoplasma strains with 100% homology. Homology analysis indicated that the CmPXXXII-hn phytoplasma strains were closely related to the phytoplasma strains, such as *Trema tomentosa* WB-hn phytoplasma strain (MW138004), with 100% homology. The CmPXXXII-hn phytoplasma strain shared 99.8% homology with the Periwinkle virescence-Malaysia strain (EU371934), 99.3% with the Yellow dwarf coconut-Malaysia strain (EU498727) and 99.4% with the Oil palm-Malaysia strain (EU498728). The results of the homology analysis of these phytoplasma strains are listed in Table 2. The analysis of single-nucleotide polymorphisms (SNP) indicated that there were many base variations and single-nucleotide insertions/deletions in the 16S rRNA gene sequences among the phytoplasma strains belonging to the 16SrXXXII group. Based on the putative consensus sequence, namely, the 16S rRNA gene sequence (MW138004) of the *Trema tomentosa* WB-hn phytoplasma strain with the length of 1303 bp, the polymorphic site positions of the analyzed genes in the tested 16SrXXXII group phytoplasma strains are numbered and listed in Table 2. Statistical analysis demonstrated that there were 16 variable loci in the 16S rRNA gene sequences of the tested 16SrXXXII group strains, and among these variable loci, two base insertions/deletions were identified, accounting for 12.5% (Table 2). Multiple sequence alignment analysis indicated that the *Candidatus* Liberibacter asiaticus strains identified in the study shared 100% homology with the *Candidatus* Liberibacter asiaticus strains that had been deposited in GenBank with the accession numbers of CP010804, MK142763 and MG384791, and so on.

### 3.4. Phylogenetic Analysis of the Plant Pathogens Identified in the Study

The phylogenetic analysis that was based on the 16S rRNA gene fragments of the phytoplasma and the *Candidatus* Liberibacter asiaticus strains indicated that the phytoplasma strains of CmPII-hn, *Tephrosia purpurea* WB-hn, *Melochia corchorifolia* WB-hnda, *Emilia sonchifolia* WB-hnda, Peanut WB-tw and Cassava WB-Vietnam (KM280679) were clustered into one clade. All the phytoplasma strains in this clade belonged to the 16SrII phytoplasma group and the bootstrap value of the clade obtained in the phylogenetic analysis was 99%. The phylogenetic tree of the plant pathogens is shown in Figure 4. From the phylogenetic tree, it can be seen that the phytoplasma strains of CmPXXXII-hn, *Trema tomentosa* WB-hn, Periwinkle virescence-Malaysia, Yellow dwarf coconut-Malaysia and Oil palm-Malaysia were clustered into one clade, with all the strains in the clade belonging to the 16SrXXXII phytoplasma group, with a 99% bootstrap value (Figure 4). As shown in Figure 4, the phytoplasma strains of CmPXXXII-hn and *Trema tomentosa* WB-hn were further divided into an independent cluster, with a 68% bootstrap value among this clade. Based on the phylogenetic analysis and the tree, the strains of *Candidatus* Liberibacter asiaticus, identified in the study, and the strains of the pathogens that had been deposited in GenBank with the accession numbers of CP010804, MK142763 and MG384791, were clustered into one independent clade, with a 99% bootstrap value (Figure 4).

## 4. Discussion

### 4.1. Citrus Maxima Single or Mixed Infection by the Phytoplasma and Candidatus Liberibacter Asiaticus

Many plants could be infected by phytoplasmas belonging to different groups [27,29,30]. *Trema tomentosa*, showing witches’ broom disease, was infected by the phytoplasma strains belonging to the 16SrXXXII-D subgroup [29]; *Melochia corchorifolia* witches’ broom disease was caused by the phytoplasma strains belonging to the 16SrII group [27]; *Waltheria indica* virescence disease was associated with the 16SrI group phytoplasma strains [30]; areca palm yellow leaf disease was also induced by the phytoplasmas belonging to the 16SrI group [38,39]. Citrus Huanglongbing disease symptoms in citrus plants, such as *Citrus maxima*, *Citrus sinensis* and *Citrus*×*limon*, were also found on the island and were identified to be associated with the pathogen of *Candidatus* Liberibacter asiaticus [23]. In the study, the *Citrus maxima* plants showing Huanglongbing symptoms were found to be associated with phytoplasmas belonging to the 16SrII-V subgroup and 16SrXXXII-D subgroup. Furthermore, to our knowledge, this is the first report in which *Citrus maxima* plants were co-infected by the phytoplasma strains belonging to the 16SrII-V subgroup and strains of *Candidatus* Liberibacter asiaticus on the island and in China, accounting for 7.4%. The findings of this study further reveal the causative agents of related plant diseases in China. Therefore, more targeted and effective epidemic monitoring, as well as prevention and control management of the related plant diseases, should be carried out in this way based on the above findings.

There are no previous reports on the mixed infection of 16SrII-V subgroup phytoplasma and *Candidatus* Liberibacter asiaticus on the plants of *Citrus maxima*. The diseased plants showing disease symptoms, including witches’ broom, yellowing, crinkled leaves and mottled leaves, may have been singly infected by one phytoplasma or co-infected by phytoplasmas belonging to different groups or phytoplasma with *Candidatus* Liberibacter asiaticus [5,25,26,40]. Sun et al. [40] reported previously that the jujube tree could be mixed infected by the phytoplasma strains belonging to the 16SrI and 16SrV groups in China. The citrus plants with Huanglongbing symptoms could be co-infected by the phytoplasma strains belonging to the 16SrIX group and *Candidatus* Liberibacter asiaticus [25], or by phytoplasma belonging to the 16SrI subgroup and *Candidatus* Liberibacter asiaticus [26,41]. In this study, the phytoplasma strains belonging to the 16SrII-V subgroup and the *Candidatus* Liberibacter asiaticus strains were simultaneously detected and identified in the *Citrus maxima* plant samples with Huanglongbing disease symptoms. However, the relationships between the different 16Sr group phytoplasmas or between the phytoplasma and the *Candidatus* Liberibacter asiaticus that simultaneously existed in the diseased plant remain unknown. Therefore, in view of the related diseases, it is highly necessary to clarify the diversity of pathogens, their primary and secondary relationships and their relationship with the occurrence of the disease symptoms—specifically, which interaction style is employed between the phytoplasma and phytoplasma or between the phytoplasma and other pathogens in the process of disease occurrence. This would be beneficial to reveal the occurrence mechanism of the related diseases, propose targeted control strategies, and improve the control effect and efficiency of related diseases.

### 4.2. The Plant Host Diversity of the Phytoplasmas and the Epidemiology of the Related Phytoplasma Diseases

The diversity of the plant hosts of the phytoplasmas is comparatively abundant [6]. Different plant hosts could be infected by phytoplasmas belonging to the same 16Sr group. The plants of chinaberry, lettuce and mulberry could be infected by the phytoplasma strains belonging to the 16SrI group [14]. The plants of *Melochia corchorifolia*, *Tephrosia purpurea* and *Emilia sonchifolia* could be infected by the phytoplasma strains belonging to the 16SrII group [27,28,42]. From this study, it could be seen that the plants of *Citrus maxima* and *Trema tomentosa* could be infected by the phytoplasma strains belonging to the 16SrXXXII-D subgroup [29]. Moreover, the same plant host could be infected by the phytoplasma strains belonging to different 16Sr groups, which would improve the spread of related diseases. It has been reported that *Melochia corchorifolia* can be infected by the phytoplasma strains belonging to the 16SrII-V and 16SrI-B subgroups [27,43]. Witches’ broom symptoms of *Tephrosia purpurea* could be induced by 16SrII-V and 16SrII-M subgroup phytoplasmas [42]. The phytoplasma strains belonging to different 16Sr groups, including 16SrI, 16SrIV, 16SrXI, 16SrXIV, 16SrXXII and 16SrXXXII, were associated with lethal yellow-type diseases in coconut [5,44,45]. Areca palm yellow leaf disease could be induced by the phytoplasma strains belonging to the 16SrI, 16SrXI and 16SrXIV groups [38,39,46,47]. The phytoplasma strains associated with papaya diseases belonged to six distinctly classified 16Sr groups containing 16SrI, 16SrII, 16SrXII, 16SrXIII, 16SrXV and 16SrVII [48].

The plant host diversity of the phytoplasmas might be conducive to the pathogens’ adaptation to different living and ecological environments, and these plant hosts of the phytoplasmas could be used as natural reservoirs or transmission vectors to assist in and promote the spread of the phytoplasmas and the epidemiology of the related diseases [14,49,50]. It was reported that paulownia witches’ broom phytoplasmas belonging to the 16SrI-D subgroup were detected among seven species of plants, including *Eleusine indica*, *Capsicum annuum*, *Setaria viridis*, *Dioscorea opposite*, *Physalis angulate*, *Arachis hypogaea* and *Cucurbita moschata*, which were probable natural hosts of the paulownia witches’ broom phytoplasmas, promoting the spread of the phytoplasmas and the epidemiology of the paulownia witches’ broom disease in Shandong Province in China [49]. Lin et al. found that the chinaberry tree (*Melia azedarach*) might serve as an alternative host of the lettuce chlorotic leaf rot phytoplasma belonging to the 16SrI-B subgroup, associated with the lettuce chlorotic leaf rot disease, which was carried by the leafhopper, *Macrosteles striifrons* [50]. The areca palm yellow leaf diseases occurring in Hainan Island in China, caused by the phytoplasmas belonging to the 16SrI group, might be spread using host plants, such as *Pericampylus glaucus*, *Waltheria indica*, pepper (*Capsicum annuum*), chinaberry and periwinkle as the pathogen transmission vectors, but not as insect vectors [8,9,39]. Through this study, it could be seen that the phytoplasma strains belonging to the 16SrXXXII-D subgroup could be transmitted through the plant hosts of *Citrus maxima* and *Trema tomentosa*, promoting the spread of the related diseases [29]. Therefore, the molecular detection, identification and diversity analysis of the phytoplasmas and the *Candidatus* Liberibacter asiaticus strains associated with *Citrus maxima* diseases are of great significance for the monitoring, prevention and control of disease in this plant species.

## 5. Conclusions

It is very important for the epidemiological monitoring, as well as the prevention and control of plant diseases, to identify clearly the pathogens of the related plant diseases and fully reveal the genetic variation of the pathogens. In this study, plants of *Citrus maxima* showing yellowing and mottled leaf symptoms on Hainan Island in China were identified to be infected by two subgroups (16SrII-V and 16SrXXXII-D) of phytoplasma and *Candidatus* Liberibacter asiaticus, and to be co-infected by 16SrII-V subgroup phytoplasma and *Candidatus* Liberibacter asiaticus. There were no variable loci detected in the 16S rRNA genes of the identified 16SrII-V subgroup phytoplasmas and the *Candidatus* Liberibacter asiaticus strains in the study. Sixteen variable loci were detected in the 16S rRNA genes of the tested 16SrXXXII group phytoplasma strains, of which base insertions/deletions accounted for 12.5% (two bases). Phytoplasmas and their plant hosts, as well as the insect vectors, are rich in diversity, and the interactions among them are comparatively complex. Therefore, the types and diversity of the pathogens clarified in the research would benefit from explaining the original occurrence, transmission vectors and distribution routes of the pathogens, demonstrating the complex interactions among the phytoplasmas, their plant hosts and their insect vectors. This would contribute to confirming the role of alternative host species in the epidemiology of the plant diseases caused by the phytoplasmas and to the development of efficient management programs for the related plant diseases.

## Figures and Tables

**Figure 1 biology-11-00869-f001:**
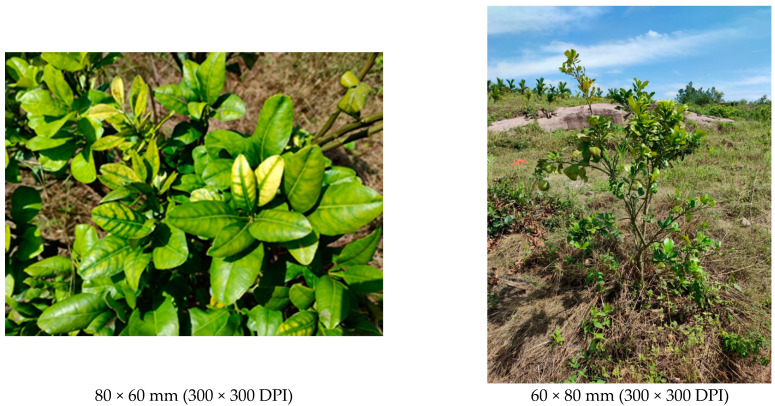
The disease symptoms of *Citrus maxima* that occur on Hainan Island in China.

**Figure 2 biology-11-00869-f002:**
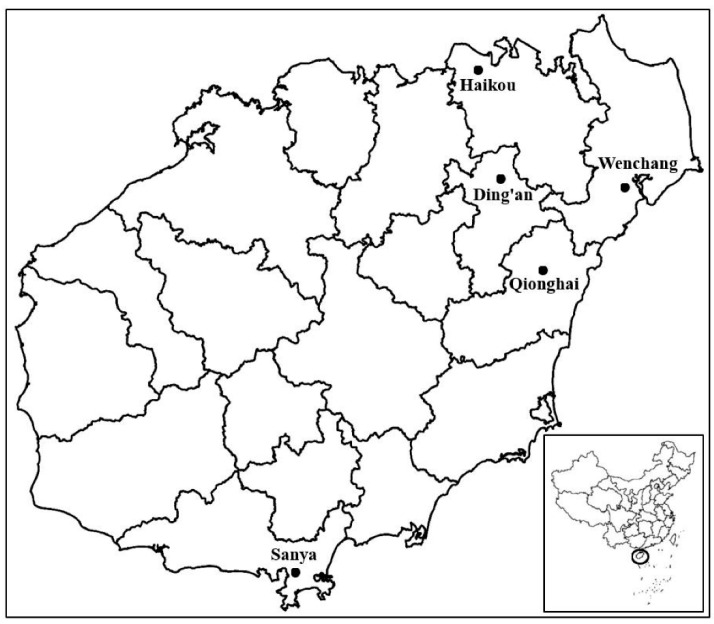
The sampling site and related locations in Hainan Province, a tropical island off the southern coast of China (sketch map).

**Figure 3 biology-11-00869-f003:**
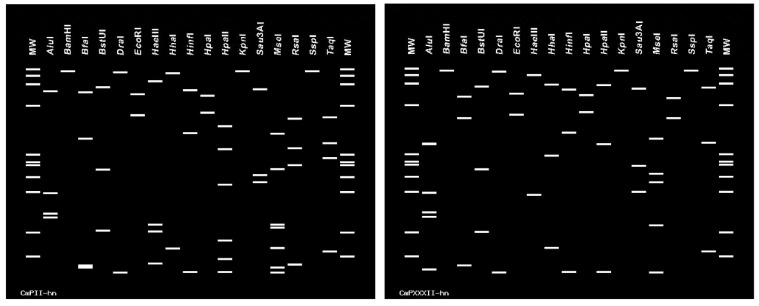
Virtual RFLP profiles of 16S rRNA gene F2nR2 fragments of the *Citrus maxima* phytoplasma CmPII-hn and CmPXXXII-hn strains. The virtual RFLP patterns of the CmPII-hn and CmPXXXII-hn phytoplasma strains are based on computer-simulated digestions of 17 restriction enzymes, including *Alu*I, *BamH*I, *Bfa*I, *Bst*UI, *Dra*I, *Eco*RI, *Hae*III, *Hha*I, *Hinf*I, *Hpa*I, *Hpa*II, *Kpn*I, *Sau*3AI, *Mse*I, *Rsa*I, *Ssp*I and *Taq*I.

**Figure 4 biology-11-00869-f004:**
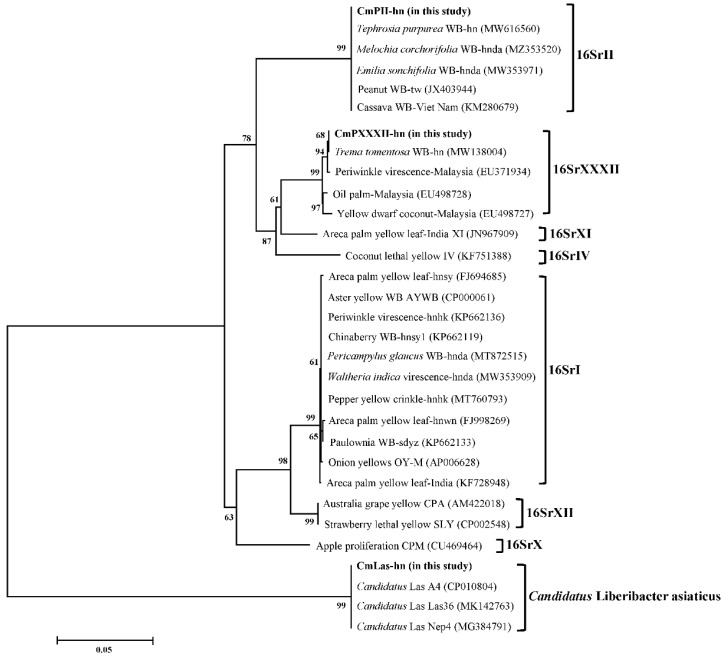
Phylogenetic tree constructed based on the 16S rRNA gene sequences of the phytoplasma and the *Candidatus* Liberibacter asiaticus strains employing the neighbor-joining method. Scale bar length represents inferred character-state changes. Branch lengths are proportional to the number of inferred character-state transformations. The percentage of replicate trees in which the associated taxa clustered together in the bootstrap test (1000 replicates) is shown next to the branches.

**Table 1 biology-11-00869-t001:** Polymorphic nucleotide sites in the 16S rRNA gene fragments of the phytoplasma strains belonging to the 16SrXXXII group.

**Phytoplasma Strains**	**Subgroups**	**Polymorphic Nucleotide Sites**															
															1	1	1
							1	1	3	6	8	8	8	9	1	2	2
		3	3	3	3	4	1	3	4	8	1	2	2	6	8	6	7
		6	7	8	9	2	5	0	6	6	8	1	9	7	3	3	2
*Trema tomentosa* WB-hn (MW138004)	16SrXXXII-D	C	A	T	T	C	A	T	T	A	G	T	T	C	A	T	C
CmPXXXII-hn (in this study)	16SrXXXII-D	•	•	•	•	•	•	•	•	•	•	•	•	•	•	•	•
Periwinkle virescence-Malaysia (EU371934)	16SrXXXII-A	•	•	•	•	•	G	•	•	G	•	•	•	-	G	•	•
Yellow dwarf coconut-Malaysia (EU498727)	16SrXXXII-B	T	G	A	C	T	G	•	•	•	•	A	C	•	•	-	G
Oil palm-Malaysia (EU498728)	16SrXXXII-C	•	G	A	C	G	G	C	C	•	A	•	•	•	•	•	G

The 16S rRNA gene sequence of the *Trema tomentosa* WB-hn phytoplasma strain (MW138004) with the length of 1303 bp was used as a consensus sequence. Only those that differ from the nucleotide in the consensus sequence are displayed. Dots (•) indicate identical nucleotides and dashes (**-**) indicate a single-nucleotide deletion. Nucleotide positions are numbered in vertical format according to the position of the first nucleotide sequence (MW138004).

**Table 2 biology-11-00869-t002:** Homology analysis of different phytoplasma strains (%).

Phytoplasma Strains	1	2	3	4	5	6	7	8	9	10	11	12	13	14	15	16	17	18	19	20	21	22
1 CmPII-hn (in this study)	100																					
2 CmPXXXII-hn (in this study)	90.6	100																				
3 *Tephrosia purpurea* WB-hn (MW616560)	100	90.6	100																			
4 *Melochia corchorifolia* WB-hnda (MZ353520)	100	90.6	100	100																		
5 Pepper yellow crinkle-hnhk (MT760793)	90.0	90.7	90.0	90.0	100																	
6 *Waltheria indica* virescence-hnda (MW353909)	90.0	90.7	90.0	90.0	100	100																
7 *Emilia sonchifolia* WB-hnda (MW353971)	100	90.6	100	100	90.0	90.0	100															
8 *Pericampylus glaucus* WB-hnda (MT872515)	90.0	90.7	90.0	90.0	100	100	90.0	100														
9 Areca palm yellow leaf-hnwn (FJ998269)	90.0	90.6	90.0	90.0	99.8	99.8	90.0	99.8	100													
10 Coconut lethal yellow IV (KF751388)	91.4	93.6	91.4	91.4	90.2	90.2	91.4	90.2	90.1	100												
11 Peanut WB-tw (JX403944)	100	90.6	100	100	90.0	90.0	100	90.0	90.0	91.4	100											
12 Chinaberry WB-hnsy1 (KP662119)	90.0	90.7	90.0	90.0	100	100	90.0	100	99.8	90.2	90.0	100										
13 Periwinkle virescence-hnhk (KP662136)	90.0	90.7	90.0	90.0	100	100	90.0	100	99.8	90.2	90.0	100	100									
14 *Trema tomentosa* WB-hn (MW138004)	90.6	100	90.6	90.6	90.7	90.7	90.6	90.7	90.6	93.6	90.6	90.7	90.7	100								
15 Periwinkle virescence-Malaysia (EU371934)	90.5	99.8	90.5	90.5	90.7	90.7	90.5	90.7	90.5	93.5	90.5	90.7	90.7	99.8	100							
16 Yellow dwarf coconut-Malaysia (EU498727)	90.1	99.3	90.1	90.1	90.7	90.7	90.1	90.7	90.5	93.1	90.1	90.7	90.7	99.3	99.2	100						
17 Oil palm-Malaysia (EU498728)	90.2	99.4	90.2	90.2	90.7	90.7	90.2	90.7	90.6	93.2	90.2	90.7	90.7	99.4	99.3	99.4	100					
18 Onion yellows OY-M (AP006628)	89.9	90.8	89.9	89.9	99.9	99.9	89.9	99.9	99.8	90.1	89.9	99.9	99.9	90.8	90.7	90.7	90.8	100				
19 Aster yellow WB AYWB (CP000061)	90.0	90.7	90.0	90.0	100	100	90.0	100	99.8	90.2	90.0	100	100	90.7	90.6	90.6	90.7	99.9	100			
20 Australia grape yellow CPA (AM422018)	89.6	90.6	89.6	89.6	96.6	96.6	89.6	96.6	96.5	89.8	89.6	96.6	96.6	90.6	90.4	90.4	90.5	96.5	96.6	100		
21 Apple proliferation CPM (CU469464)	89.9	91.4	89.9	89.9	92.9	92.9	89.9	92.9	92.7	91.5	89.9	92.9	92.9	91.4	91.2	91.2	91.3	92.8	92.9	92.9	100	
22 Strawberry lethal yellow SLY (CP002548)	89.6	90.6	89.6	89.6	96.6	96.6	89.6	96.6	96.5	89.8	89.6	96.6	96.6	90.6	90.4	90.4	90.5	96.5	96.6	100	92.9	100

## Data Availability

The DNA sequences of the assessed genes are available in the GenBank database, and the accession numbers are given in the paper. All other relevant data are provided within the paper.

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
