# Peer review of "Molecular Identification and Characterization of Two Groups of Phytoplasma and *Candidatus* Liberibacter Asiaticus in Single or Mixed Infection of *Citrus maxima* on Hainan Island of China"

_biology, 2022, doi:10.3390/biology11060869_

Round 1
Reviewer 1 Report
Comments to the Author
It is important to identify the pathogens associated with abnormal symptoms including yellowing and mottled leaves in Citrus maxima in Hainan Island of China. Authors analyzed and compared genetic variation and phylogenetic relationship of the pathogens based on 16S rRNA and β‐operon gene fragments specific for phytoplasma and Candidatus Liberibacter asiaticus. This work laid the foundation for further research focused on Phytoplasmas and Candidatus Liberibacter asiaticus Infecting Citrus maxima.
My main concern is that
1.There are already many reports the pathogens associated with abnormal symptoms including yellowing and mottled leaves in Citrus maxima. What is the main finding of this manuscript? Authors should highlight it by add some sentences or reorganized the manuscript structure.
2.Authors should reconsider the title, it should more focus on the information provided in text.
3.Authors should check figure and text more carefully, there are some mistakes (see below).
Other comments:
- In Fig 1 & 2, where is the scale?
- In line 265 “Phylogenetic tree was constructed by the neighbor‐joining method using …” have show in line 146,
- there are four times of “To our knowledge, this is the first report that the Citrus maxima plants were infected separately by the 16SrII‐V subgroup phytoplasma and the 16SrXXXII‐D subgroup phytoplasma in China, and this is also the first report that the Citrus maxima plants were mixed infected by the 16SrII‐V subgroup phytoplasma and the Candidatus Liberibacter asiaticus in China”, please try to use different word.
Author Response
Reviewer 1
Comments to the Author
It is important to identify the pathogens associated with abnormal symptoms including yellowing and mottled leaves in Citrus maxima in Hainan Island of China. Authors analyzed and compared genetic variation and phylogenetic relationship of the pathogens based on 16S rRNA and β‐operon gene fragments specific for phytoplasma and Candidatus Liberibacter asiaticus. This work laid the foundation for further research focused on Phytoplasmas and Candidatus Liberibacter asiaticus Infecting Citrus maxima.
Response: Thank you for the reviewer’s careful comments. According to the reviewer’s comments, the revisions were made in the manuscript and highlighted in blue, the detailed reply for the revisions were as follows.
My main concern is that
1.There are already many reports the pathogens associated with abnormal symptoms including yellowing and mottled leaves in Citrus maxima. What is the main finding of this manuscript? Authors should highlight it by add some sentences or reorganized the manuscript structure.
Response: According to the reviewer’s comments, the main findings of the manuscript were highlighted and some sentences were reorganized. The revisions were highlighted in blue.
2.Authors should reconsider the title, it should more focus on the information provided in text.
Response: The title of the text was reorganized and revised in the manuscript. The revisions were highlighted in blue.
3.Authors should check figure and text more carefully, there are some mistakes (see below).
Response: Some mistakes in the figures and text were checked and revised. The revisions were highlighted in blue.
Other comments:
- In Fig 1 & 2, where is the scale?
Response: According to the reviewer’s comment, the scale was added in Fig 1 indicating the size of picture. The revision was highlighted in blue. Fig 2 is a sketch map, indicating the general location of the sampling and related sites, Hainan Island of China and so on, no relationship with the size and distance of the map, therefore, the authors did not list the scale.
- In line 265 “Phylogenetic tree was constructed by the neighbor‐joining method using …” have show in line 146,
Response: The sentences were reorganized and revised in the manuscript. The revisions were highlighted in blue.
- there are four times of “To our knowledge, this is the first report that the Citrus maxima plants were infected separately by the 16SrII‐V subgroup phytoplasma and the 16SrXXXII‐D subgroup phytoplasma in China, and this is also the first report that the Citrus maxima plants were mixed infected by the 16SrII‐V subgroup phytoplasma and the Candidatus Liberibacter asiaticus in China”, please try to use different word.
Response: The sentences were reorganized and revised in the manuscript. The revisions were highlighted in blue.
Reviewer 2 Report
Dear Authors
The present study focuses on Molecular Identification and Characterization of Phytoplasmas and Candidatus Liberibacter asiaticus Infecting Citrus maxima
in Hainan Island of China and claims that the first report in its category. The manuscript have several opportunities for further improvements, please find my suggestions here
- Simple summary and abstract seems to be overlapping, please reorganize the abstract in clearer way to demonstrate the key findings.
- Line 42-43, please avoid writing that it is the first report.
- Line 95- Please explain the “abnormal symptoms” what does it mean?
- Line 101- Control samples represents the healthy leaves of same plant? Or other plants in similar locations?
- Line 102-105, Better suits for discussion.
- Please provide the details of weather conditions during the sampling times, if possible?
- Line 178-188, Please reframe the sentence, it is too long and not clear.
- Any effort was done to isolate the pure cultures from infected leaves to check the morphological differences of bacterial colonies?
- Line 299-301, seems like repetition of similar claims many times throughout the results and discussion, please check and reframe it.
- Conclusion may be more focused on the key findings and their importance for future researchers.
Thank you
Author Response
Reviewer 2
Dear Authors
The present study focuses on Molecular Identification and Characterization of Phytoplasmas and Candidatus Liberibacter asiaticus Infecting Citrus maxima
in Hainan Island of China and claims that the first report in its category. The manuscript have several opportunities for further improvements, please find my suggestions here
Response: Thank you for the reviewer’s careful comments. According to the reviewer’s comments, the revisions were made in the manuscript and highlighted in blue, the detailed reply for the revisions were as follows.
- Simple summary and abstract seems to be overlapping, please reorganize the abstract in clearer way to demonstrate the key findings.
Response: The simple summary and abstract were rewritten and revised in the manuscript. The revisions were highlighted in blue.
- Line 42-43, please avoid writing that it is the first report.
Response: The sentences were reorganized and revised in the manuscript. The revisions were highlighted in blue.
- Line 95- Please explain the “abnormal symptoms” what does it mean?
Response: Abnormal symptoms means the diseased symptoms, the words and the sentences were revised in the manuscript. The revisions were highlighted in blue.
- Line 101- Control samples represents the healthy leaves of same plant? Or other plants in similar locations?
Response: Yes, control samples represents the healthy leaves of same plant from the same location. The sentence in the manuscript was rewritten as “the healthy Citrus maxima samples were also collected in the same locations as control”.
- Line 102-105, Better suits for discussion.
Response: In order to describe the sampling location information more detailed, these sentences were added in this part. And the sentences were revised in the manuscript. The revisions were highlighted in blue.
- Please provide the details of weather conditions during the sampling times, if possible?
Response: For detection of the pathogens in the study maybe not affected by the weather conditions. Therefore, we did not record the details of weather conditions during the sampling times. All the samples were collected on sunny days by us. We will pay attention to weather conditions during the sampling times in the future.
- Line 178-188, Please reframe the sentence, it is too long and not clear.
Response: The sentences were reorganized and revised in the manuscript. The revisions were highlighted in blue.
- Any effort was done to isolate the pure cultures from infected leaves to check the morphological differences of bacterial colonies?
Response: In our study, only phytoplasmas and Candidatus Liberibacter asiaticus were detected according to the previous study and the symptoms. These pathogens are inability to be cultured in vitro.
- Line 299-301, seems like repetition of similar claims many times throughout the results and discussion, please check and reframe it.
Response: These sentences were reorganized and revised throughout the full text, and the revisions in the manuscript were highlighted in blue.
- Conclusion may be more focused on the key findings and their importance for future researchers.
Response: The conclusion of the manuscript was reorganized by the authors and the revisions were highlighted in blue.
Reviewer 3 Report
The MS of Yu et al. entitled “Molecular identification and characterization of phytoplasmas and Candidatus Liberibacter asiaticus infecting Citrus maxima in Hainan Island of China” contributes to the better understanding of the complexity of infection carried out by phytoplasma in conjunction to other bacteria in pomelo (Citrus maxima). Overall, the paper is very hard to read because of the very poor English writing. In several sections, almost every second sentence needs revisions. It would be nice for authors to mention common name of Citrus maxima as well. The introduction is very short and needs to be totally rewritten. The material and methods provide all the info needed; however, some info regarding the phylogenetic analyses have to be moved from the results section 3.4 in this section. Discussion is OK but the conclusion section has to be rewritten as is too similar to the simple summary and the abstract. Overall, the approach was straightforward, but the amount of work and its significance is not impressing. In the past, it was not possible to report research requiring such a rather limited amount of work in a journal with an IF of over 5. Also, if I am wrong, 25% of the citations are self citations. I would say that this is very high even if the main author has significant contributions in working with phytoplasmas.
Specific, selected comments
Abstract
Lines 29-31
“The results indicated that the pathogens including phytoplasma strains CmPII-hn belonging to 16SrII-V subgroup and CmPXXXII-hn belonging to 16SrXXXII-D subgroup, identified by iPhyClassifier, Candidatus Liberibacter asiaticus strains CmLas-hn were detected in the diseased plant samples, with numbers of 12, 2 and 6 out of 54 respectively.”
This is a very confusing sentence that needs to be rewritten.
Lines 40-42
“The Candidatus Liberibacter asiaticus strains identified in the study and deposited in the GenBank such as the strain (CP010804) and so on were in one independent cluster with 99 % bootstrap value.”
This sentence is not well written and hard to understand its meaning. It needs to be rewritten.
Lines 42-44
“To our knowledge, this is the first report that Citrus maxima was infected separately by 16SrII‐V subgroup phytoplasmas, 16SrXXXII‐D subgroup phytoplasmas.”
The sentence incorrect and has to be revised.
Lines 44-47
“And this is also the first report that the plant was mixed infected by the 16SrII‐V subgroup phytoplasmas and the Candidatus Liberibacter asiaticus in China. It would provide a theoretical foundation and be benefit for the epidemic monitoring and the effective prevention and control of the related diseases.”
This is not the first report of mixed infection. It is of local interest to mention that it was observed for the first time in China. Also, if mixed infections were reported elsewhere, why this study would provide “theoretical foundation”?
Introduction
Lines 57-59
“For phytoplasmas are inability to be cultured in vitro, molecular analyses of DNA sequences from these organisms are more important for their detection, identification and classification.”
This sentence is poorly written. It has to be corrected.
Lines 59-61
“The 16S rRNA gene sequence, served as the standards for identification of bacterial species, is mainly used to molecular identification and characterization of phytoplasmas.”
Sentence should be written better.
Lines 63-65
“Based on the 16S rRNA and few other conserved gene sequences from several hundred phytoplasma strains all over the world, 28 groups of phytoplasmas had been classified [7].”
Sentence is hard to understand, it has to be rewritten.
Line 71
“were mixed infected”
Find alternative wording.
Lines 76-80
“The previous research results indicated that the pathogens of Citrus Huanglongbing disease in China were only the Candidatus Liberibacter asiaticus, and there were no other Candidatus Liberibacter spp. such as Candidatus Liberibacter africanus and Candidatus Liberibacter americanus were detected in the plant samples with Citrus Huanglongbing diseases.”
Reformulate the sentence, it is not clear what the authors want to say.
Lines 87-92.
“Therefore, the detection and identification of the phytoplasma and Candidatus Liberibacter spp. from the diseased plant samples of Citrus maxima with abnormal symptoms including yellowing and mottled leaves were simultaneously carried out in this study, which provided a scientific basis and a theoretical foundation for clarifying the pathogens associated with the diseased symptoms of the Citrus maxima host and for more targeted and effective prevention and control of such the related diseases.”
The sentence is extremely long, hard to understand, with several mistakes (for example: ”of such the related diseases” ?!?). The sentence has to be broken in at least 2 different sentences.
Materials and methods
Lines 98-99
What kind of “ecological damage”?
Line 107
What “according to the methods of CTAB”? There is only one method!
Lines 141-142
It is awkward to say “analyzed to learn biological information of the DNA fragments and the pathogen”. Why “biological information”? DNA sequence was used to ID the pathogen!
Author Response
Reviewer 3
Comments and Suggestions for Authors
The MS of Yu et al. entitled “Molecular identification and characterization of phytoplasmas and Candidatus Liberibacter asiaticus infecting Citrus maxima in Hainan Island of China” contributes to the better understanding of the complexity of infection carried out by phytoplasma in conjunction to other bacteria in pomelo (Citrus maxima). Overall, the paper is very hard to read because of the very poor English writing. In several sections, almost every second sentence needs revisions. It would be nice for authors to mention common name of Citrus maxima as well. The introduction is very short and needs to be totally rewritten. The material and methods provide all the info needed; however, some info regarding the phylogenetic analyses have to be moved from the results section 3.4 in this section. Discussion is OK but the conclusion section has to be rewritten as is too similar to the simple summary and the abstract. Overall, the approach was straightforward, but the amount of work and its significance is not impressing. In the past, it was not possible to report research requiring such a rather limited amount of work in a journal with an IF of over 5. Also, if I am wrong, 25% of the citations are self citations. I would say that this is very high even if the main author has significant contributions in working with phytoplasmas.
Response: Thank you for the reviewer’s careful comments. According to the reviewer’s comments, the revisions were made in the manuscript and highlighted in blue, the detailed reply for the revisions were as follows.
Specific, selected comments
Abstract
Lines 29-31
“The results indicated that the pathogens including phytoplasma strains CmPII-hn belonging to 16SrII-V subgroup and CmPXXXII-hn belonging to 16SrXXXII-D subgroup, identified by iPhyClassifier, Candidatus Liberibacter asiaticus strains CmLas-hn were detected in the diseased plant samples, with numbers of 12, 2 and 6 out of 54 respectively.”
This is a very confusing sentence that needs to be rewritten.
Response: The sentences were rewritten in the manuscript and the revisions were highlighted in blue.
Lines 40-42
“The Candidatus Liberibacter asiaticus strains identified in the study and deposited in the GenBank such as the strain (CP010804) and so on were in one independent cluster with 99 % bootstrap value.”
This sentence is not well written and hard to understand its meaning. It needs to be rewritten.
Response: The sentences were rewritten in the manuscript and the revisions were highlighted in blue.
Lines 42-44
“To our knowledge, this is the first report that Citrus maxima was infected separately by 16SrII‐V subgroup phytoplasmas, 16SrXXXII‐D subgroup phytoplasmas.”
The sentence incorrect and has to be revised.
Response: The sentences were rewritten in the manuscript and the revisions were highlighted in blue.
Lines 44-47
“And this is also the first report that the plant was mixed infected by the 16SrII‐V subgroup phytoplasmas and the Candidatus Liberibacter asiaticus in China. It would provide a theoretical foundation and be benefit for the epidemic monitoring and the effective prevention and control of the related diseases.”
This is not the first report of mixed infection. It is of local interest to mention that it was observed for the first time in China. Also, if mixed infections were reported elsewhere, why this study would provide “theoretical foundation”?
Response: Yes, this is the first report that Citrus maxima can be infected by 16SrII-V and16SrXXXII-D subgroups phytoplasmas in China. And this is also the first report that the plants are co-infected by 16SrII-V subgroup phytoplasmas and Candidatus Liberibacter asiaticus in China. “theoretical foundation” was deleted in the manuscript and the sentences were rewritten. The revisions were highlighted in blue.
Introduction
Lines 57-59
“For phytoplasmas are inability to be cultured in vitro, molecular analyses of DNA sequences from these organisms are more important for their detection, identification and classification.”
This sentence is poorly written. It has to be corrected.
Response: The sentences were rewritten in the manuscript and the revisions were highlighted in blue.
Lines 59-61
“The 16S rRNA gene sequence, served as the standards for identification of bacterial species, is mainly used to molecular identification and characterization of phytoplasmas.”
Sentence should be written better.
Response: The sentences were rewritten in the manuscript and the revisions were highlighted in blue.
Lines 63-65
“Based on the 16S rRNA and few other conserved gene sequences from several hundred phytoplasma strains all over the world, 28 groups of phytoplasmas had been classified [7].”
Sentence is hard to understand, it has to be rewritten.
Response: The sentences were rewritten in the manuscript and the revisions were highlighted in blue.
Line 71
“were mixed infected”
Find alternative wording.
Response: The words were rewritten throughout the full text and the revisions were highlighted in blue.
Lines 76-80
“The previous research results indicated that the pathogens of Citrus Huanglongbing disease in China were only the Candidatus Liberibacter asiaticus, and there were no other Candidatus Liberibacter spp. such as Candidatus Liberibacter africanus and Candidatus Liberibacter americanus were detected in the plant samples with Citrus Huanglongbing diseases.”
Reformulate the sentence, it is not clear what the authors want to say.
Response: The sentences were rewritten in the manuscript and the revisions were highlighted in blue.
Lines 87-92.
“Therefore, the detection and identification of the phytoplasma and Candidatus Liberibacter spp. from the diseased plant samples of Citrus maxima with abnormal symptoms including yellowing and mottled leaves were simultaneously carried out in this study, which provided a scientific basis and a theoretical foundation for clarifying the pathogens associated with the diseased symptoms of the Citrus maxima host and for more targeted and effective prevention and control of such the related diseases.”
The sentence is extremely long, hard to understand, with several mistakes (for example: ”of such the related diseases” ?!?). The sentence has to be broken in at least 2 different sentences.
Response: The sentences were rewritten in the manuscript and the revisions were highlighted in blue.
Materials and methods
Lines 98-99
What kind of “ecological damage”?
Response: The sentences were rewritten in the manuscript and the revisions were highlighted in blue.
Line 107
What “according to the methods of CTAB”? There is only one method!
Response: The sentences were rewritten in the manuscript and the revisions were highlighted in blue.
Lines 141-142
It is awkward to say “analyzed to learn biological information of the DNA fragments and the pathogen”. Why “biological information”? DNA sequence was used to ID the pathogen!
Response: The sentences were rewritten in the manuscript and the revisions were highlighted in blue.
Round 2
Reviewer 1 Report
Extensive editing of English language and style required
Author Response
Dear reviewer
Thank you for the reviewer’s careful comments. According to the reviewer’s comments, English editing was made in our manuscript using a professional language institution from MDPI.
We would be happy to make any further changes that may be required. Thank you for your consideration and we look forward to hearing from you.
Sincerely,
Shaoshuai Yu
Chinese Academy of Tropical Agricultural Sciences

Reviewer 2 Report
Dear Authors
Thank you for answering all the queries, i do not have any further questions.
Regards
Author Response
Dear reviewer
Thank you for your careful and valuable comments. According to the editor’s and the reviewers’ comments, English editing was made in our manuscript using a professional language institution from MDPI.
Sincerely,
Shaoshuai Yu
Chinese Academy of Tropical Agricultural Sciences

Reviewer 3 Report
The authors re-wrote large sections of the MS. Some of the information added improved the MS. Most of the new information, unfortunately, is written in such a way that makes reading the MS an unpleasant experience. This MS badly needs some serious polishing done by a person with decent proficiency in English language. I encourage the authors to seek for professional help to improve English language throughout the MS. Also, there are many places in the MS in which information provided is redundant.
Below I provide just a few examples of sentences that needs re-writing:
Title:
Molecular Identification and Characterization of Two Groups 2 Phytoplasmas and Candidatus Liberibacter asiaticus Single/Mixed Infecting Citrus maxima in Hainan Island of China
The title in not written using proper English.
Maybe it can be something like:
Molecular Identification and Characterization of Two Groups 2 Phytoplasmas and of Candidatus Liberibacter asiaticus in Single or Mixed Infected Citrus maxima in Hainan Island of China
Line 13
Why “detected results”? Can results be detected and undetected?
Line 22
“Diseased” should be replaced with “disease”. “Diseased symptoms” is nonsense!
Lines 22 and 23
The sentence including “were found that could be infected separately … and could be mixed infected”
Is not OK.
It should be something like:
were found to be either infected separately … or were mixed infected
Lines 29-31
The results indicated that the pathogens in including phytoplasma strains ….. were identified in the diseased plant samples
Should be changed to something like:
“ Pathogens such as phytoplasma strains ….….. were identified in the diseased plant samples.”
Line 41
“Bad” should be “have”.
Line 43
“that Citrus maxima ….” should be replaced by “ showing that Citrus maxima ….”
Lines 42-46
Please make one more concise sentence from the 2 sentences pasted below. Avoid using China twice. It is obvious that the authors talk about the same location/country.
“To our knowledge, this is the first report that Citrus maxima can be infected by 16SrII-V and16SrXXXII-D subgroups phytoplasmas in China. And this is also the first report that the plants are co-infected by 16SrII-V subgroup phytoplasmas and Candidatus Liberibacter asiaticus in China.”
Lines 70
Please, do not start sentences with “and”.
For example: And for
Line 325
It is clear that the experiment was carried out in China, Hainan Island. This was repeated in the MS several times. Therefore, there is no need to repeat this again and again in the discussion section. For example: “In Hainan Island of China, many plants could be infected by the phytoplasmas belonging to different groups.” Mentioning Hainan is sufficient. Are there any other islands called Hainan that belong to other countries?
Also, in the discussion, do not focus only on what is important for China. Try to make it more general, for all areas in which Citrus maxima, and other plants from the family Rutaceae can grow.
Author Response
Dear reviewer
Thank you for your careful and valuable comments. According to your comments, English editing was made in our manuscript using a professional language institution from MDPI. The revisions were made in the manuscript. The detailed reply for the revisions were as follows.
The authors re-wrote large sections of the MS. Some of the information added improved the MS. Most of the new information, unfortunately, is written in such a way that makes reading the MS an unpleasant experience. This MS badly needs some serious polishing done by a person with decent proficiency in English language. I encourage the authors to seek for professional help to improve English language throughout the MS. Also, there are many places in the MS in which information provided is redundant.
Below I provide just a few examples of sentences that needs re-writing:
Title:
Molecular Identification and Characterization of Two Groups 2 Phytoplasmas and Candidatus Liberibacter asiaticus Single/Mixed Infecting Citrus maxima in Hainan Island of China
The title in not written using proper English.
Maybe it can be something like:
Molecular Identification and Characterization of Two Groups 2 Phytoplasmas and of Candidatus Liberibacter asiaticus in Single or Mixed Infected Citrus maxima in Hainan Island of China
Response: According to the reviewer’s comments, the title of the manuscript was rewritten.
Line 13
Why “detected results”? Can results be detected and undetected?
Response: The sentences were rewritten in the manuscript.
Line 22
“Diseased” should be replaced with “disease”. “Diseased symptoms” is nonsense!
Response: The word was rewritten in the manuscript.
Lines 22 and 23
The sentence including “were found that could be infected separately … and could be mixed infected”
Is not OK.
It should be something like:
were found to be either infected separately … or were mixed infected
Response: The sentences were rewritten in the manuscript.
Lines 29-31
The results indicated that the pathogens in including phytoplasma strains ….. were identified in the diseased plant samples
Should be changed to something like:
“ Pathogens such as phytoplasma strains ….….. were identified in the diseased plant samples.”
Response: The sentences were rewritten in the manuscript.
Line 41
“Bad” should be “have”.
Response: The words and the sentences were rewritten in the manuscript.
Line 43
“that Citrus maxima ….” should be replaced by “ showing that Citrus maxima ….”
Response: The sentences were rewritten in the manuscript.
Lines 42-46
Please make one more concise sentence from the 2 sentences pasted below. Avoid using China twice. It is obvious that the authors talk about the same location/country.
“To our knowledge, this is the first report that Citrus maxima can be infected by 16SrII-V and16SrXXXII-D subgroups phytoplasmas in China. And this is also the first report that the plants are co-infected by 16SrII-V subgroup phytoplasmas and Candidatus Liberibacter asiaticus in China.”
Response: The sentences were rewritten in the manuscript.
Lines 70
Please, do not start sentences with “and”.
For example: And for
Response: The words and sentences were rewritten throughout the full text.
Line 325
It is clear that the experiment was carried out in China, Hainan Island. This was repeated in the MS several times. Therefore, there is no need to repeat this again and again in the discussion section. For example: “In Hainan Island of China, many plants could be infected by the phytoplasmas belonging to different groups.” Mentioning Hainan is sufficient. Are there any other islands called Hainan that belong to other countries?
Response: According to the reviewer’s comments, these words and sentences were reorganized and revised throughout the full text.
Also, in the discussion, do not focus only on what is important for China. Try to make it more general, for all areas in which Citrus maxima, and other plants from the family Rutaceae can grow.
Response: The revisions were made in the discussion of the manuscript.
We would be happy to make any further changes that may be required. Thank you for your consideration and we look forward to hearing from you.
Sincerely,
Shaoshuai Yu
Chinese Academy of Tropical Agricultural Sciences

Round 3
Reviewer 3 Report
Though is some places the MS could be further improved, the current version is acceptable.